# 3D Swin Transformer for Partial Medical Auto Segmentation

Aneesh Rangnekar[0000−0002−0079−9495], Jue Jiang, Harini Veeraraghavan

Memorial Sloan Kettering Cancer Center
rangnea@mskcc.org

**Abstract.** Transformers are the highest accuracy segmentation frameworks in computer vision for natural imagery from the past few years. In contrast, medical imaging approaches, except a select few (for example, SwinUNETR and SMIT), are still dominated by the nnU-Net architecture family. In this paper, we investigate the application of a hierarchical vision transformer to the FLARE-23 challenge.

Specifically, we benchmark our results using a relatively lightweight architecture, Swin-X Seg. We use multi-model self-training, wherein we use nnU-Net for predicting pseudo labels on partially labeled cases and then optimize the transformer architecture for memory requirements. Our network achieved the average DSC scores of 83.13 % and 35.19 % on the open validation set (50 cases) for organs and tumors, respectively, while staying under a max GPU memory utilization of 4GB at evaluation runtime. Our results show that there is potential for the transformer architecture to perform at par or better than conventional convolutional approaches, and we hope our findings encourage more research in the area.

**Keywords:** Auto Segmentation · Self-training · Swin Transformer.

## 1 Introduction

Accurate, fast, and automated volumetric segmentation of organs and tumors is essential for radiotherapy treatment planning. It often constitutes one of the time-consuming parts of radiation treatment planning workflows [37]. Abdominal organs are particularly time-consuming to segment owing to the presence of a large number of organs as well as due to the random and large variation in the appearance and shape of gastrointestinal organs and limited soft-tissue contrast on clinically used computed tomography (CT) images. Hence, deep learning methods to generate segmentation are under active development [20,2].

Deep learning methods have shown the capability to generate multi-organ segmentation for abdomen [16,18,34,1] and other disease sites. The availability of well-curated public challenge datasets [20,2] has enabled the evaluation of various methods using the same reference benchmark with well-defined metrics. However, a fundamental prerequisite of well-curated pixel-wise annotations or volumetric segmentations of the various organs for training these networks must

be more expensive and time-consuming to generate on large datasets. One recent promising approach to alleviate the need for large, curated datasets is the self-supervised pretraining followed by a fine-tuning approach that has demonstrated success in medical image analysis, mainly when using transformer-based architectures[34,18]. Swin UNETR [34] and SMIT [18] have shown that using self-supervised learning (SSL) improves the performance of transformer-based networks on semantic segmentation, as compared to training the networks from scratch. Our approach builds on these methods and utilizes a transformer architecture [21] for segmentation with a pretraining step (self-supervised learning) using labeled and unlabeled examples followed by fine-tuning.

We also follow the FLARE-23 rules, whereby, unlike prior works[34,18], which used a large number of CT scans from various disease sites for pretraining, we used only the 4,000 example scans provided as part of the training set for self-supervised pretraining. Furthermore, keeping with the requirements for using a relatively small architecture with limited memory requirements, we also constructed a lightweight transformer architecture.

Our learning framework uses multi-model self-training [41,42,32], where the teacher is an fine-tuned nnU-Net [15] that generates pseudo labels for the various categories. The student network uses a Swin transformer backbone [21] segmentation network (here on referred to as Swin-X Seg) that accepts a combination of FLARE-23 and pseudo labeled examples for fine-tuning (Fig. 1). Our initial studies show that naively using the partially labeled dataset, with a transformer backbone to obtain pseudo labels, results in poor performance across multiple categories [36,5,40]. . Hence, we resort to this combination of semi-supervised learning, wherein the teacher is an nnU-Net and the student is Swin-X Seg.

Our approach allows us to fully utilize the partially-labeled training dataset to its fullest extent, while leveraging fundamental augmentation techniques shown to be effective in natural image analysis. This mitigates the need for requiring complex approaches like the CutMix [43] or ClassMix [29], wherein extensive registration would be required before mixing two 3D scans so that the networks do not lose understanding of organ placements, especially with architectures that rely heavily on positional information.

Our key contributions are (a) a lightweight 3D vision transformer applied to multi-organ and tumor segmentation, (b) the SSL approach extending prior works by learning the downstream task using partial labels, and the application of this approach on an open-source FLARE-23 dataset.

## 2    Method

### 2.1    Overview

We studied the performance of hierarchical vision transformer-based U-Net architecture on the FLARE-23 challenge. Vision transformers require large amounts of data [36,5,40,19] to achieve high generalization performance. Hence, FLARE-23, which consists of 4,000 training images, provides a nice test bed for evaluating

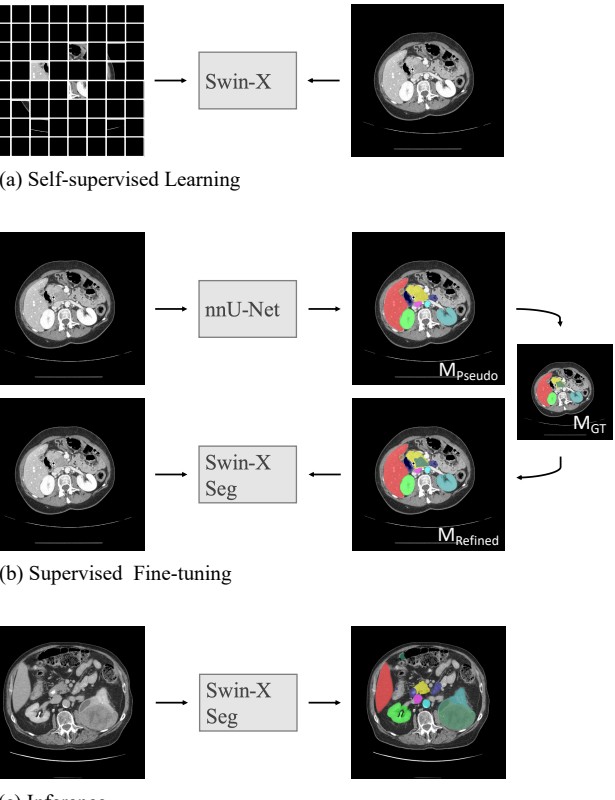

**Fig. 1.** Our three-stage pipeline: (a) self-supervised training of the backbone network [17], (b) uses a combination of pseudo labels ($M_{Pseudo}$) [15] and FLARE-23 provided annotations ($M_{GT}$) to obtain refined labels ($M_{Refined}$) for learning segmentation, and (c) inference on a new unseen volumetric scan.

vision transformer architectures. However, 1800 CTs in FLARE-23 are unlabeled with the remaining 2200 CTs provided with partial labels, wherein some but not all the 14 different organs and tumors were segmented, which makes supervised training challenging. Therefore, we used a two-step training approach consisting of: (i) self-supervised pretraining performed on the entire dataset of 4,000 CTs without using any segmentations for supervised training, and (ii) supervised fine-tuning that combined fully labeled CTs together with CTs with pseudo labels created using a different model. We discuss each part of our approach in detail, and the specificities involved in our final implementation.

## 2.2 Preprocessing

We used the following preprocessing steps in all our experiments:

– Reorient the scans to the right-anterior-superior (RAS) view.

– Clip the intensities based on the Hounsfield units to [-250, 250].
– We resize all scans to $x, y, z$ volumetric spacings of $1.0, 1.0, 1.0$ during training and inference.
– In addition, we randomly sample 4 scans of $96 \times 96 \times 96$ size from each scan as training examples, representing 2 positive and 2 negative samples for the network at every instance.

### 2.3   Proposed Method

**Choice of Transformer:**
Hierarchical Vision Transformers [21,8] are pyramid-shaped architectures that rely on gradual down-sampling, similar to convolutional neural networks, while maintaining a global look-out with their multi-scale designs. We use the Swin-Transformer backbone for our approach as it has been widely adopted for 3D medical auto segmentation [34,18] and shown to be more accurate than the vanilla vision transformer[7].

Swin UNETR [34] and SMIT [18] have over 60 million (M) parameters. Whereas Swin UNETR processes data at $96 \times 96 \times 96$, SMIT processes data at $128 \times 128 \times 128$ resolution. Both methods use sliding windows for generating final inference. The FLARE-23 constraints require memory efficient inference. A straightforward memory efficient approach to reduce the total number of flops used for inference would be to utilize CT scans reduced to $96 \times 96 \times 96$ pixels, at the risk of decreasing the image resolution, which can impact accuracy for smaller organs. Hence, we reduced the number of parameters used in the network by decreasing the total number of blocks per depth to the final $2 - 2 - 2 - 2$ configuration as well as reduced the total number of channels through the UNETR architecture using $1 \times 1$ convolutions. This reduced the network size from 60M parameters to 31M parameters, a relatively lightweight architecture compared to current state-of-the-art methods. This is also crucial towards keeping the GPU requirements under 4GB as stipulated under FLARE-23 rules.

**Self-supervised Learning:** The SSL approach made use of the self-distillation based pretext tasks used in the SMIT [18], including namely Masked Image Modeling (MIM), Masked Patch self-Distillation (MPD) and Image Token self-Distillation (ITD). SMIT performs self distillation by concurrently maintaining an online teacher model ($NET_T$) with the same network architecture as the student model ($NET_S$) [35]. The loss functions used to optimize the network are briefly discussed here and we refer interested details to the original paper[18] for more details.

Suppose $\{x_1, x_2\}$ are two augmented views of a 3D image $x$. $N$ image patches are extracted from the images to create a sequence of image tokens [7]. The image tokens are then corrupted by randomly masking image tokens based on a binary vector, with a probability $p$, and then replacing with mask token [3]. The second augmented view $v$ is also corrupted but using a different mask vector instance. In this order, the three losses deal with the views in the following manner:

- **Masked Image Prediction (MIP)** $\rightarrow x_1$, $NET_S$, involves dense pixel regression of image intensities within masked patches using the context of unmasked patches [12].
- **Masked patch token self-distillation (MPD):** $\rightarrow x_1$, $NET_S$, $NET_T$, trains the student network to predicts the tokens of the teacher network (distillation).
- **Global image token self-distillation (ITD):** $\rightarrow x_1, x_2$, $NET_S$, $NET_T$, learns to match the global image embedding of the view-scan seen by the student network to the view-scan seen by the teacher network.

SSL training is performed by optimizing the network using all three aforementioned losses. FLARE-23 rules dictate that no external data be used. Hence, following the rules, SSL used the same 4,000 CTs provided as part of the training set. No segmentations provided with the data was used for network optimization in this step.

**Supervised Fine-tuning:**

In order to fully utilize all available training data to improve accuracy, we used the best performing nnU-Net model, the winner from FLARE22[15] to provide pseudo labels for the partially labeled and unlabeled datasets the FLARE 23 training sets. We only use 735 examples from the 2200 images that contain a labeled instance of tumor, with the combination of FLARE-23 and nnU-Net pseudo labels (Fig. 1). We trained our network sing a combination of Dice loss and cross-entropy loss following previous approaches [24,16,34,18].

### 2.4 Post-processing

No data specific post processing was used following pixel-level classifications generated by the segmentation methods. Sliding window inference with 50% overlap was used for generating segmentations for the whole 3D image volumes.

## 3 Experiments

### 3.1 Dataset and evaluation measures

The FLARE-23 challenge is an extension of the FLARE 2021-2022 [26][27], aiming to promote the development of foundation models in abdominal disease analysis. The segmentation targets cover 13 organs and various abdominal lesions around the organs. The dataset comprises scans from more than 30 medical centers, including TCIA [6], LiTS [4], MSD [33], KiTS [13,14], autoPET [10,9], TotalSegmentator [39], and AbdomenCT-1K [28], with appropriate licensing. The training set includes 4,000 abdomen CT scans, 2,200 CT scans with partial segmentation labels for some of them, and 1,800 CT scans without any segmentation labels. The validation and testing sets include 100 and 400 CT scans, respectively, covering various abdominal cancer types, such as liver, kidney, pancreas, colon, and gastric, to name a few. The organ annotation process used ITK-SNAP [44], nnU-Net [16], and MedSAM [25].

**Table 1.** Development environments and requirements.

| System | Ubuntu 18.04.5 LTS |
|---|---|
| CPU | AMD EPYC 7543P 32-Core Processor @ 2.8 Ghz |
| RAM | 128 GB |
| GPU (number and type) | NVIDIA A100 80 GB × 4 |
| CUDA version | 11.8 |
| Programming language | Python 3.8 |
| Deep learning framework | Pytorch 1.13 ± CUDA 11.7 [30] |
| Specific dependencies | MONAI, SimpleITK, Nibabel |
| Code | https://github.com/The-Veeraraghavan-Lab/FLARE23 |

**Table 2.** Training protocols.

| Network initialization | SSL-FLARE-23 [18] |
|---|---|
| Batch size | 4 |
| Patch size | 96 × 96 × 96 |
| Total epochs | 100 |
| Optimizer | AdamW [23] |
| Initial learning rate (lr) | 2e-4 |
| Lr decay schedule | Linear Warmup with Cosine Annealing [22,11] |
| Training time | 33 hours |
| Loss function | Cross-Entropy Loss /w Dice Loss |

The evaluation metrics encompass two accuracy measures—Dice Similarity Coefficient (DSC) and Normalized Surface Dice (NSD)—alongside two efficiency measures—running time and instantaneous GPU maximum memory consumption.

### 3.2  Implementation details

**Environment settings** The development environments and requirements are presented in Table 1. We provide all the requirements in our released codebase on GitHub.

**Training protocols** The model training protocols are shown in in Table 2. An image patch size of 96 × 96 × 96 with random 3D flips performed on the data to provide augmented samples was used for network training.

**Table 3.** Quantitative evaluation results. Segmentation accuracy results (DSC and NSD with mean and standard deviation) are reported on the publicly provided 50 validation cases made available by the FLARE-23 organizers.

| Target | Public Validation | |
| --- | --- | --- |
| | DSC(%) | NSD(%) |
| Liver | $96.08 \pm 4.230$ | $93.58 \pm 10.66$ |
| Right Kidney | $87.00 \pm 20.81$ | $83.37 \pm 21.81$ |
| Spleen | $93.24 \pm 9.730$ | $90.92 \pm 14.23$ |
| Pancreas | $80.47 \pm 7.860$ | $89.99 \pm 7.020$ |
| Aorta | $90.55 \pm 14.80$ | $91.61 \pm 16.30$ |
| Inferior vena cava | $87.88 \pm 6.800$ | $86.97 \pm 9.300$ |
| Right adrenal gland | $77.35 \pm 17.46$ | $87.78 \pm 19.00$ |
| Left adrenal gland | $72.44 \pm 15.83$ | $82.03 \pm 16.59$ |
| Gallbladder | $75.61 \pm 28.21$ | $71.61 \pm 30.06$ |
| Esophagus | $74.81 \pm 16.56$ | $84.85 \pm 15.99$ |
| Stomach | $89.17 \pm 9.110$ | $87.60 \pm 11.85$ |
| Duodenum | $70.78 \pm 10.77$ | $84.21 \pm 9.240$ |
| Left kidney | $85.65 \pm 21.81$ | $82.33 \pm 23.22$ |
| Tumor | $35.19 \pm 30.17$ | $22.99 \pm 22.10$ |
| Average (Organ) | $83.13 \pm 8.440$ | $85.55 \pm 12.58$ |
| Average | $79.70 \pm 11.43$ | $81.08 \pm 14.93$ |

## 4 Results and discussion

### 4.1 Quantitative results on validation set

Table 3 shows our Swin-X Seg's performance on the 50 validation cases provided by the FLARE-23 organizers. The network was slightly less accurate ($< 80\%$ DSC) for organs such as the adrenal glands, gallbladder, esophagus, duodenum, as well as for tumors compared to larger organs like the liver, spleen, left and right kidneys, and the stomach. The tumor segmentation accuracy was low because of the larger variability in the types of tumors analyzed and the relatively few examples with complete labels. Overall, the network accuracy was lower for smaller organs like the adrenal glands and gallbladder when compared to larger organs like the liver. Poor accuracy for organs also resulted when they were adjacent to the tumors.

Table 4 shows that inference requirements of under 4GB GPU memory consumption were satisfied for all cases. However, all except two cases (0001, 0019) did not satisfy the running time requirement under 60 secs owing to sliding window-based inference, with 50% overlap. A natural option is to use sliding window inference without any overlap (0%). However, this results in a poor overall score (77% DSC average on organ, 27% DSC on tumor); hence, we did

**Table 4.** Quantitative evaluation of segmentation efficiency of the reported cases using running time and maximum GPU memory consumption ($< 4096$ MB). Evaluation GPU platform: A100 (80GB).

| Case ID | Image Size | Running Time (s) | Max GPU (MB) |
|---------|-----------|------------------|--------------|
| 0001 | (512, 512, 55) | 28.01 | 3464 |
| 0051 | (512, 512, 100) | 65.86 | 3850 |
| 0017 | (512, 512, 150) | 73.94 | 3896 |
| 0019 | (512, 512, 215) | 48.00 | 3616 |
| 0099 | (512, 512, 334) | 69.28 | 3756 |
| 0063 | (512, 512, 448) | 84.76 | 3776 |
| 0048 | (512, 512, 499) | 74.73 | 3748 |
| 0029 | (512, 512, 554) | 102.5 | 4032 |

not pursue it. In addition, we optimized for test-time efficiency by performing foreground thresholding to use only the body regions for analysis by ignoring the surrounding air for inference. Our analysis showed that in cases with larger field of view, wherein the body occupied higher volume the inference time utilization increased (e.g. 0017 > 0019, 0063 > 0048).

### 4.2 Qualitative results on validation set

Figures 2 and 3 show the segmentations generated by our network on representative examples taken from the validation set of FLARE-23. As shown in Fig.2, whereas the model tends to consistently segment the normal tissues with high accuracy, misclassifications occur within tumor regions, tumor voxels classified as the kidney, despite achieving a relatively high DSC accuracy for the tumors. The higher DSC accuracy for tumors is not surprising given the larger tumor volumes. On the other hand, as shown in Fig. 3 for really large tumors such as #0057 and #0095, the algorithm generated highly inaccurate segmentation, misclassifying the tumors occurring on the left side of anatomy as liver. #0027 shows an example where the kidney tumor was correctly segmented together with the kidney adjacent to the tumor, although the esophagus occurring distally to the pancreatic head was misclassified as pancreas. Similarly, in #0089, the pancreas is oversegmented by the model, whereas the kidney tumor encased within the kidney is undersegmented, highlighting the challenges, particularly when the tumor and the healthy tissues are adjacent to each other.

### 4.3 Segmentation efficiency results on validation set

We optimized for segmentation inference efficiency by extracting the foreground or the body as a preprocessing step using standard image thresholding. No additional optimization was performed in terms of training or testing. Even this simple approach showed that it is possible to improve inference efficiency as seen in Table 4.

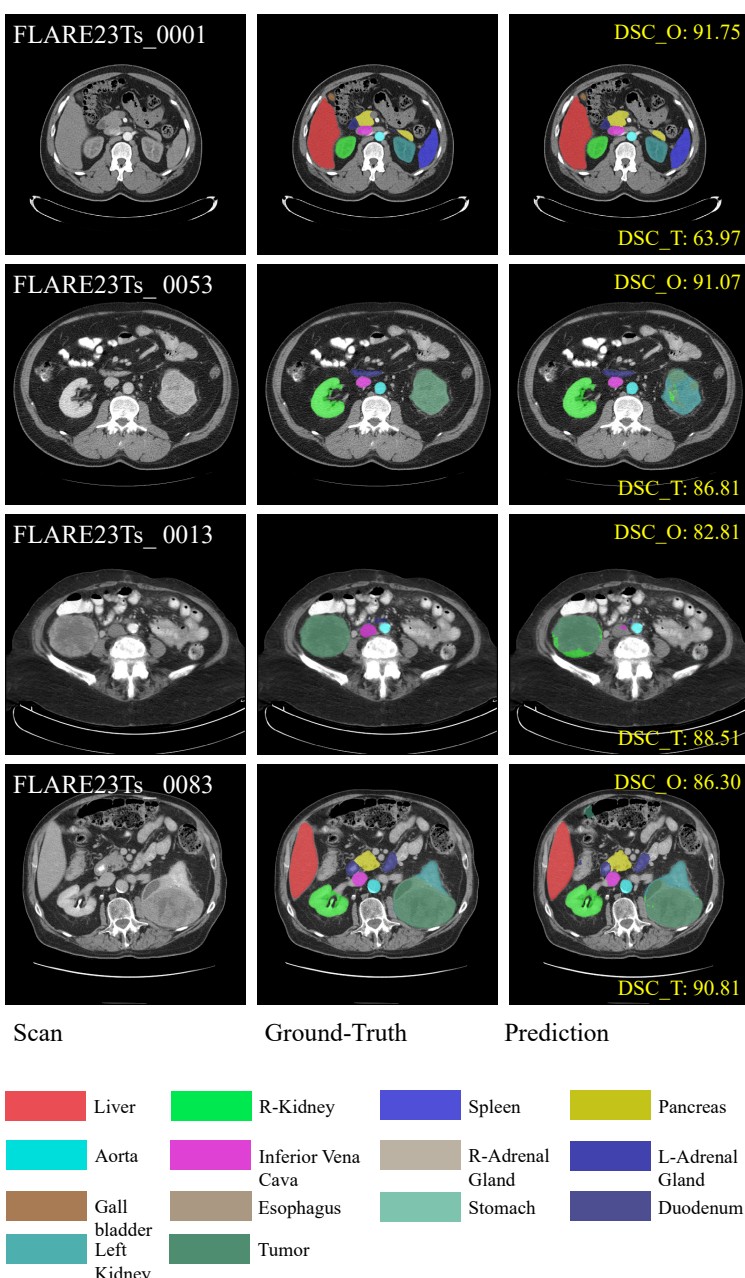

**Fig. 2.** Example scans showing relatively good performance in terms of misclassifications by the trained Swin-X Seg model. DSC_T refers to tumor DSC and DSC_O refers to average multi-organs DSC.

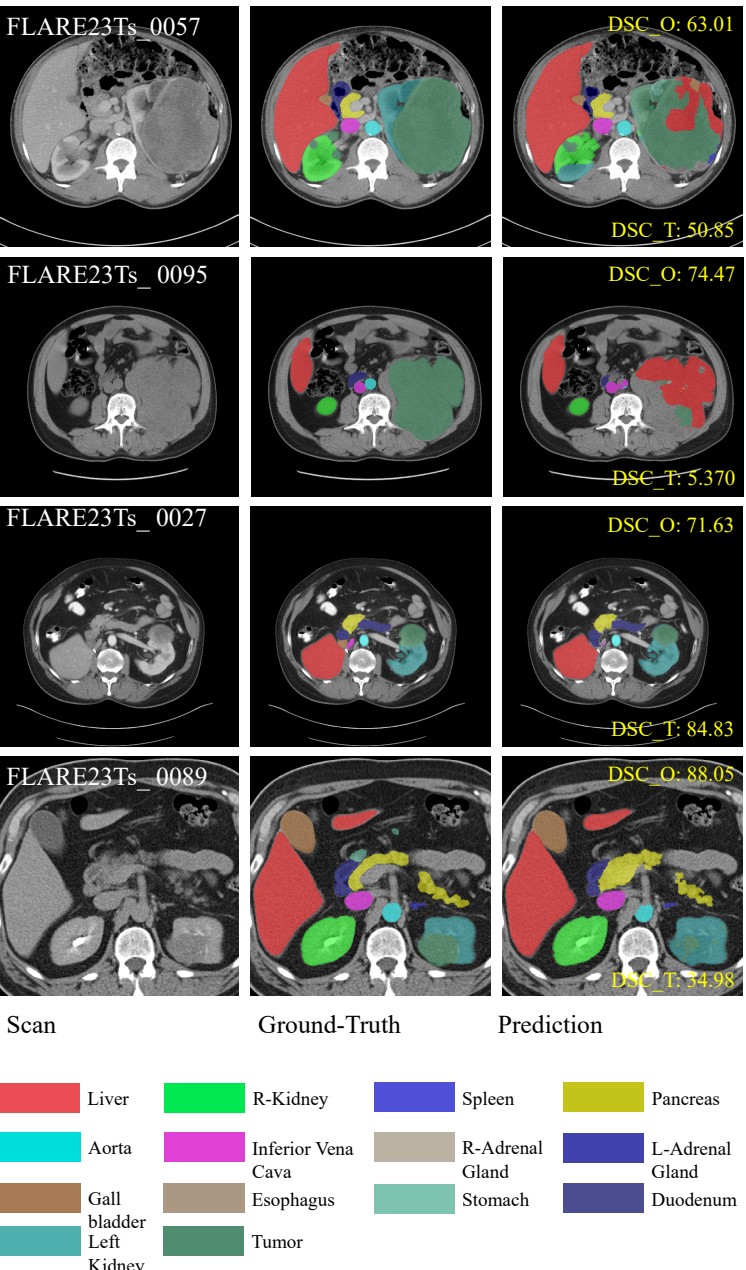

**Fig. 3.** Example scans showing relatively poor performance in terms of misclassifications by the trained Swin-X Seg network. DSC_T refers to tumor DSC and DSC_O refers to average multi-organs DSC.

### 4.4   Results on final testing set

This is a placeholder. We will send you the testing results during MICCAI (2023.10.8). (This is to be left as is.)

### 4.5   Limitation and future work

Our goal was to evaluate the capability of transformer-based approach for multi-organ and tumor segmentation. We used a relatively lightweight (31M) in order to satisfy the memory requirements of the competition as well as to study to what extent such methods are successful in comparison to convolutional-based approaches such as the nnU-Net used in the previous iteration of the competition [15,38]. Our approach to use nnU-Net generated pseudo labels was motivated by prior results using Semiformer [40], which showed poor accuracy with vision transformer with small labeled training samples can be improved when combined with pseudo labels produced by convolutional neural networks (CNN). However, VITs have generally shown to be more accurate than CNN models. Hence, one approach is to use VIT instead of a CNN for providing pseudo labels. its important to note that the approach combining pseudo labels with CNN and larger VIT models becomes impractical due to increasing memory needs. Another limitation of our approach is the poor segmentations we observed on the tumor and tissue interface, which we plan to address in the future.

## 5   Conclusion

We presented our approach, multi-model self-training, that used nnU-Net to generate pseudo labels and then Swin transformer to establish a foundation for research into auto segmentation with pseudo labels. In addition, we also identify limitations and discuss research approaches to mitigate them, including knowledge distillation and semi-supervised learning. We believe that our framework serves as a good foundation for further research into efficient network designs and methodology for accurate medical image segmentation.

**Acknowledgements** The authors of this paper declare that the segmentation method they implemented for participation in the FLARE-23 challenge has not used any pre-trained models and additional datasets other than those provided by the organizers. The proposed solution is fully automatic without any manual intervention. We thank all the data owners for making the CT scans publicly available and CodaLab [31] for hosting the challenge platform. This research was partly funded through grant from NCI R01CA258821-01A1 and the Memorial Sloan Kettering (MSK) Cancer Center Support Grant/Core Grant NCI P30 CA008748.

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

**Table 5.** Checklist Table. Please fill out this checklist table in the answer column.

| Requirements | Answer |
|---|---|
| A meaningful title | Yes |
| The number of authors ($\leq 6$) | 3 |
| Author affiliations and ORCID | Yes |
| Corresponding author email is presented | Yes |
| Validation scores are presented in the abstract | Yes |
| Introduction includes at least three parts: background, related work, and motivation | Yes |
| A pipeline/network figure is provided | Fig. 1 |
| Pre-processing | Pages 3, 4 |
| Strategies to use the partial label | Page 5 |
| Strategies to use the unlabeled images | Page 5 |
| Strategies to improve model inference | Page 4 |
| Post-processing | Pages 5, 6 |
| Dataset and evaluation metric section is presented | Page 5 |
| Environment setting table is provided | Table 1 |
| Training protocol table is provided | Table 2 |
| Ablation study | N/A |
| Efficiency evaluation results are provided | Table 4 |
| Visualized segmentation example is provided | Figures 2, 3 |
| Limitation and future work are presented | Yes |
| Reference format is consistent. | Yes |