# OpenReview forum: "3D Swin Transformer for Partial Medical Auto Segmentation"
_MICCAI.org/2023/FLARE — Submitted to FLARE 2023_

### Official Review · Reviewer_AfHz · 2023-09-27
**Swin-transformer for medical segmentation with self-supervised and pseudo-label training**

**Rating:** 8
**Confidence:** 5

**Review:**

The author used a light swin transformer (swin-x) to improve the computation efficiency. The model is trained with self-supervised pretraining on the entire dataset of 4,000 CTs. The author also used nn-unet to generate pseudo-labels for the final finetuning. The paper is clear and well written.

---

### Official Review · Reviewer_obhB · 2023-10-02
**Transformer based segmentation**

**Rating:** 7
**Confidence:** 4

**Review:**

The paper describes authors Flare23 challenge submission, with a swin transformer based segmentation network.  The architecture was adjusted to fit within the compute requirements.  The paper follows the provided template well, and includes all the experimental results details.

---

### Official Review · Reviewer_H1GW · 2023-10-04
**The 3D Swin Transformer was appropriate.**

**Rating:** 7
**Confidence:** 4

**Review:**

The manuscript details the authors' submission for the Flare23 challenge, utilizing a Swin Transformer-based segmentation network. Adjustments were made to the architecture to align with computational constraints. The paper adheres well to the given template and encompasses all details of the experimental results. However, Table 3 should not only include publication validation but also Online Validation and Testing

---

### Public Comment · ~PENGJU_LYU1 · 2023-11-26
**add test results**

there are still online validation and test results lacking

---

### Decision · Program_Chairs · 2023-10-24

Accept